# Effect of Ozonation on the Mechanical, Chemical, and Microbiological Properties of Organically Grown Red Currant (*Ribes rubrum* L.) Fruit

**DOI:** 10.3390/molecules27238231

**Published:** 2022-11-25

**Authors:** Piotr Kuźniar, Justyna Belcar, Miłosz Zardzewiały, Oskar Basara, Józef Gorzelany

**Affiliations:** 1Department of Food and Agriculture Production Engineering, University of Rzeszow, 4 Zelwerowicza Street, 35-601 Rzeszów, Poland; 2Doctoral School of the University of Rzeszów, University of Rzeszów, st Rejtana 16C, 35-959 Rzeszów, Poland

**Keywords:** red currant fruit, organic cultivation, ozonation, mechanical properties, bioactive compounds, antioxidant activity, microbiological stability

## Abstract

Red currant fruits are a valuable source of micro- and macronutrients, vitamins, and chemical compounds with health-promoting properties, the properties of which change depending on the harvest date and the time and method of storage. This study analysed the effect of applying 10 ppm ozone gas for 15 and 30 min on the mechanical properties, chemical properties and microbiological stability of three organic-grown red currant fruit cultivars. Fruits harvested at the time of harvest maturity had significantly larger diameters and weights and lower water contents compared with fruits harvested seven days earlier, and the ozonation process, regardless of its harvesting date, reduced the physical parameters in question (diameter, weight, and water content). The ascorbic acid content of the ozonated fruit varied, with the highest decreases observed for fruit harvested 7 days before the optimal harvest date and stored for 15 days under refrigeration (an average decrease of 13.31% compared with the control fruit without ozonation). In general, the ozonation process had a positive effect on the variation of fruit antioxidant activity, with the highest average values obtained for fruit harvested 7 days before the optimum harvest date and stored for 15 days under refrigeration conditions; in addition, it also had an effect on reducing the development of microorganisms, including mesophilic aerobic bacteria, yeasts, and moulds, mainly for the cultivar ‘Losan’.

## 1. Introduction

Red currant (*Ribes rubrum* L.) is a shrub widely cultivated in Eastern Europe (Russia, and Ukraine) and Central Europe (Poland, Germany, and France) in both commodity plantations and home gardens. The popularity of red currant cultivation is due to the ease of establishing and maintaining plantations [1]. The yield of red currant depends significantly on the cultivar, the size of the bushes, the age of the shoots, the environmental conditions during their growing season, the cultivation system, and the time of ripening [2,3]. Stress on red currant plants caused by abiotic factors, for example, water shortages or too low or high average daily air temperature during the growing season, significantly affects the chemical composition of the fruit, including the content of ascorbic acid or polyphenolic compounds [4]. The mechanical harvesting of red currant fruit not only results in a significant reduction in the costs associated with manual harvesting but also in the speed and efficiency of the combined harvest conducted, which is influenced by the method of plantation management, the size of the bushes, the height of berries in the crown, and the selection of varieties with an even ripening time and the relative ease of fruit detachment [5]. Mechanical damage (including abrasion, bruising, or crushing) occurs during the harvesting, transport, and processing of the fruit, which may even result in the elimination of a batch of raw material from the market [6].

Three currant species characterised by black (*Ribes nigrum* L.), red (*Ribes rubrum* L.), and white-cream skin colouring (*Ribes niveum* L.) are cultivated in Poland and are differentiated both in terms of taste and chemical composition [7]. Red currant fruits are used for direct consumption, but also in many branches of food processing. Due to their rich chemical composition, red currant fruits are used for juices, jams, jellies, frozen foods, ice creams, desserts, wines, and liqueurs [7,8,9]. Red currant fruits contain sugars (mainly glucose and fructose), organic acids (including malic and citric acids), and polyphenolic compounds, including flavonoids or anthocyanins, micro- and macronutrients (Ca, Zn, Mg, P, Fe, Cu, Mn), vitamins, and fibre, the content of which depends both on the cultivar and abiotic conditions during the growing season [1,7,8,10]. Red currant fruit, compared with black currant, is characterised by significantly lower ascorbic acid content, which depends on the date of harvest, the degree of ripeness, and the cultivar, most often ranging from 29.3 to 71.2 mg·100 g^−1^ f.w. [11]. Due to the high antioxidant activity of red currant fruits, they are used as a raw material for the prevention of cancer, diseases related to abnormal heart function (hypertension, atherosclerosis, ischaemic heart disease), and kidney disease [7,10].

The postharvest storage of red currant fruit not only causes a loss of turgor as a result of water loss but also alters the chemical composition of the fruit. Immediately after harvest, red currant fruit can be stored under refrigeration or subjected to a freezing process to inhibit adverse biochemical processes that occur in the fruit after harvest [12]. Ozonation can be used to reduce the quality losses of cold-stored fruit. The use of the ozonation process has a positive effect on reducing water losses during fruit storage, increases antioxidant activity, and causes a reduction in the release of ethylene from the treated fruit [13,14,15,16]. Ozone is a highly oxidizing chemical that disinfects the treated plant material, thus extending its technological shelf life [17]. Ozone can be used in two forms, aqueous or gaseous, but studies on fruit ozonation have shown better process efficiency with the latter form [13,14,16]. Ozonation technology that extends the shelf life of food is considered a nonthermal method of food preservation that improves food safety without compromising food quality or polluting the environment [12]. The disinfectant and biocidal properties of ozone have attracted the interest of the fruit and vegetable sector because the ozone molecule rapidly decomposes into oxygen, leaving no residue on the product. Ozone reacts with some organic food compounds, resulting in the formation of possible byproducts, e.g., aldehydes, ketones, and carboxylic acids, which pose no risk to human health [14,15,16,17].

The purpose of this study was to determine the feasibility of using ozonation to improve the quality of organically grown red currant fruit, including mechanical, microbiological, and chemical properties during cold storage.

## 2. Results and Discussion

### 2.1. Changes in the Mechanical Properties of Red Currant Fruit

The refrigeration of red currant fruit affects not only the water content but is also associated with changes in mechanical properties, which are important in the management and development of optimal processing technology for a given raw material. The fruits of the studied red currant cultivars, subjected to strength tests, differed in their morphological characteristics and were modified by cultivar, harvest date, storage time, and ozonation time (Table 1). The largest diameter, as well as weight, and the smallest density and moisture content were characterised by the fruits of the ‘Holenderska Czerwona’ cultivar, while, significantly, the highest density was found in the fruits of the ‘Luna’ cultivar. Fruits harvested at harvest maturity had significantly higher diameter and weight and significantly lower water content. The water content of the red currant fruits and all their analysed morphological characteristics decreased during storage. However, a significant decrease in diameter and weight values occurred after 8 days of storage and in density and water content after 15 days (Table 1). The ozonation process, regardless of its duration, reduced the weight, diameter, and water content of currant fruit (Table 1). In addition, the dose of ozone selected on the basis of preliminary tests did not cause any visible damage to the fruit epidermis. An opposite relationship, i.e., a smaller decrease in moisture content after ozonation, was observed by Zapałowska et al. [18] for sea buckthorn fruit, Zardzewiały et al. [17] for rhubarb petioles, and Gorzelany et al. [19] for stored ground cucumbers.

In fact, the fruits of the ‘Luna’ cultivar were the most resistant to mechanical damage, since they required the greatest force and energy for destruction (Table 2). Fruits of this variety also deformed the most and had the highest apparent modulus of elasticity. On the other hand, the fruits of the ‘Losan’ cultivar were the most susceptible to damage, as they were damaged with significantly less force and energy and had the lowest apparent modulus of elasticity (Table 2.). The application of ozonation decreased the values of the determined mechanical parameters of the fruits, except that the decrease was significant only for energy independent of the ozonation time. The energy and modulus of elasticity decreased significantly during the storage period, and strength decreased significantly after 8 days of storage (Table 2).

The ozonated red currant fruit, irrespective of the duration of the process, showed a higher apparent modulus of elasticity during storage; i.e., it had better elastic properties (Figure 1). The energy and force required to break down ozonated currant fruit were significantly lower after one day of storage than for non-ozonated fruit. However, after 15 days of storage, the fruit ozonated for 30 min had higher values of energy and destructive force; that is, they were more resistant to mechanical damage. Zapałowska et al. [18] reported a decrease in the values of destructive force and energy during storage for both ozonated and non-ozonated sea buckthorn fruits. The ozonation of sea buckthorn fruit with an ozone concentration of 10 ppm for 15 and 30 min increased the resistance to mechanical damage. On the contrary, ozonating the sea buckthorn fruit for 5 min decreased its resistance to mechanical damage. This may mean that a longer ozonation time than 30 min is necessary to increase the resistance to mechanical damage of the red currant fruit tested. Furthermore, Antos et al. [20] observed an increase in damage strength values after ozonation for apple tissue. Horvitz and Cantalejo [21] studied red pepper fruit cut in strips and ozonated with 0.7 µL·L^−1^ ozone gas for 1, 3, and 5 min after 1 and 7 days of refrigerated storage, and they found a decrease in firmness compared with the control. Since the firmness of the non-ozonated peppers decreased faster, the ozonated fruits had a slightly higher firmness than the control after 14 days of sampling.

### 2.2. Changes in pH and Acidity in Red Currant Fruit in Relation to Harvesting Date and Ozonation Time

The content of organic acids, in addition to sugars, in red currant fruit is the main determinant of its palatability and consumer acceptability. Citric acid and malic acid are the main representatives among the organic acids found in red currant fruit [1]. Figure 2 and Figure 3 show the effect of the harvest date and ozonation time on changes in pH and acidity in red currant fruit.

The red currant fruit harvested 7 days before the optimum harvest date without ozonation had a pH ranging from 3.20 to 3.27, and after 15 days of cold storage, the pH of the red currant fruit increased to 3.23–3.55. Non-ozonated fruit harvested at the optimum harvest date had an average pH that was 6.92% higher, with the highest differences observed for ‘Holenderska Czerwona’ (22.14% increase) compared with fruit harvested at an earlier date. In general, the ozonation process did not have a statistically significant effect on the change in pH of the red currant fruit harvested both a week before the optimal harvest date and at the optimum harvest date, regardless of cold storage, with the exception of the ‘Holenderska Czerwona’ cultivar. In a study by Gorzelany et al. [22], non-ozonated Saskatoon berry fruits were characterized by a pH of 4.12–5.03, and ozonation for 15 min increased the pH of the fruits by an average of 5.32%, while in a study by the same team on sea buckthorn fruits, the pH of the non-ozonated fruits was 3.02–3.19, and an ozonation process carried out for 15 min (ozone gas concentration of 10 ppm) increased the pH of sea buckthorn fruits by an average of 2.88% [23].

The acidity of red currant fruit harvested seven days before the optimal date (constituting the control sample) ranged from 0.95 to 1.09 g·100 g^−1^. After 15 days of cold storage, there was a decrease in the acidity of red currant fruit by an average of 64.00%, regardless of cultivar. The fruit harvested on the optimum harvest date had an average acidity of 0.59 g·100 g^−1^, while fruit storage resulted in an average acidity decrease of 18.15%. In a study by Djordjević et al. [1], the acidity of red currant fruit was at a level of 0.7–1.6 g·100 g^−1^, while in a study performed a decade earlier, the acidity of red currant fruit ranged from 1.0 to 1.9 g·100 g^−1^ [12]. Studies of red currant fruit acidity by Petrisor et al. [24] showed that it was higher and was at the level of 2.33–3.12 g·100 g^−1^, while in a study by Milivojević et al. [9], the acidity of red currant fruit was in the range of 0.17–0.24 g·100 g^−1^. The ozonation process of red currant fruit harvested one week before the optimal date increased acidity by an average of 11.35% for an ozonation time of 15 min and by an average of 14.09% for a time of 30 min (for the cultivars ‘Czerwona Holenderska’ and ‘Losan’), while the ozonation process of red currant fruit of the cultivar ‘Luna’ resulted in a decrease in acidity by an average of 17.89% compared with the control. For the other variants analysed, no statistically significant changes were observed in the acidity of the fruit harvested at the optimal harvest date and stored for 15 days under refrigeration and subjected to the ozonation process. In comparison, the ozonation process reduced the acidity of Saskatoon berry fruits by an average of 43.85% for fruits treated with 10 ppm gaseous ozone for 15 min and by an average of 26.39% for fruits treated with ozone for 30 min [22], and in a study of sea buckthorn fruits, the ozonation process reduced the acidity by an average of 5.26% for fruits treated with 10 ppm gaseous ozone [23]. Ozone can activate the antioxidant defence mechanism in plant cells and metabolize reactive oxygen species (ROS), which can become an important regulator of the antioxidant potential of plant cells, including acids; a shorter increase was observed in ozonated fruits [25].

### 2.3. Content of Bioactive Compounds in Red Currant Fruit

The content of ascorbic acid, a chemical compound with antioxidant properties, in red currant fruit, significantly depends on several factors, including the cultivar, harvest time or storage conditions, and the duration of the raw material [26]. Ascorbic acid is found in many fruit varieties, including those commonly found and consumed in Poland (average 65 mg·100 g^−1^), such as raspberries (average 29 mg·100 g^−1^), blackberries (average 21 mg·100 g^−1^ ascorbic acid; [26]), black currant fruit (average 205 mg·100 g^−1^), and white currant fruit (average 32 mg·100 g^−1^), as well as red currant fruit (average 41 mg·100 g^−1^ ascorbic acid; [7,11]). The ascorbic acid content of the red currant fruit harvested seven days before the optimal harvest date not subjected to ozonation ranged from 31.2 to 44.1 mg·100 g^−1^, while fruit harvested at the optimum harvest date had higher ascorbic acid content by 21.79% on average. Refrigerated storage increased the ascorbic acid content of the fruit by an average of 36.93% for the fruit harvested one week before the optimal harvest date, while it had no statistically significant effect on the ascorbic acid content of fruit harvested at the optimum harvest date (Table 3). In a study by Djordjević et al. [1], the ascorbic acid content of the red currant fruit ranged from 24.6 to 66.9 mg·100 g^−1^, while in earlier studies, the ascorbic acid content was higher, ranging from 33.4 to 71.6 mg·100 g^−1^ [12]. In a study by Petrisor et al. [24] the ascorbic acid content of red currant fruit ranged from 35.4 to 52.3 mg·100 g^−1^, while Berk et al. [10] determined that the ascorbic acid content of the fruit was between 30.16 and 38.05 mg·100 g^−1^. The red currant fruit ozonation process affected the variation in ascorbic acid content, with the highest decreases observed for fruit harvested 7 days before the optimal harvest date and stored for 15 days under refrigeration (an average decrease of 13.31% for fruit ozonated for 15 min and an average decrease of 3.4% for fruit ozonated for 30 min compared with the control sample), while the highest increase in ascorbic acid content was observed in ozonated red currant fruit harvested at the optimal harvest date and stored under refrigeration conditions (an average increase of 10.17% for fruit ozonized for 15 min and an average increase of 12.50% for fruit ozonised for 30 min compared to the control sample). The highest increases in ascorbic acid content were observed for the ozonated fruit of the red currant cv. ‘Luna’ compared with the control (Table 3). Appropriate ozonation can significantly increase peroxidase activity (POD), inhibit polyphenol oxidase (PPO) activity, maintain high levels of total phenols (TP) and flavonoids, improve the antioxidant capacity of fruit, and preserve fruit quality [15]. Ozone in strawberry fruit decreased the rate of formation of superoxide radical anions, and the content of hydrogen peroxide increased the activity of superoxidase (SOD), catalase (CAT), ascorbate peroxidase (APX), and monodehydroascorbate reductase (MDHAR), and it also promoted the accumulation of ascorbic acid (ASA) [27]. The ozonation of raspberry fruit significantly increased the activity of mitochondrial respiratory enzymes, such as succinate dehydrogenase, cytochrome C oxidase, and H +-ATPase, which contributed to maintaining a high level of ATP and energy charge in the fruit during storage. In addition, energy metabolism in mitochondria was closely correlated with the antioxidant potential of raspberry fruit. Enzymatic changes in ozonated fruit affect acid changes, including the content of ascorbic acid, which has an antioxidant effect [28].

Among bioactive compounds, polyphenols are the most abundant group of chemical compounds found in red currant fruit. Fruits harvested seven days before the optimal harvest date were characterised by a polyphenol content ranging from 117.4 to 201.7 mg GAE·100 g^−1^, and storage time reduced the parameter in question by 23.16% on average. The red currant fruits harvested on the optimum harvest date without ozonation were characterised by an average total polyphenol content of 73.27 mg GAE·100 g^−1^, while cold storage influenced their slight decrease (by 3.74% on average; Table 3). Djordjević et al. [1], studying red currant fruit, obtained a polyphenol content of 101.2 to 325.2 mg GAE·100 g^−1^, while in earlier studies, the content of the parameter in question was, on average, lower by one half, ranging from 67.2 to 153.4 mg GAE·100 g^−1^ [12]. In the study by Petrisor et al. [24], the content of the total polyphenols in the red currant fruit ranged from 95.21 to 150.35 mg GAE·100 g^−1^. Laczkó-Zöld et al. [29] obtained a total polyphenol content ranging from 72.76–192.98 mg GAE·100 g^−1^ depending on the extract used, while in the study by Jakobek et al. [30], the content of total polyphenols in red currant fruit was 194.7 mg GAE·100 g^−1^. The red currant fruit affected the content of total polyphenols; ozonation for 15 min decreased the content of the analysed parameter by 5.63% on average in relation to the control sample, regardless of the date of fruit harvest and storage time, while ozonation for 30 min had a positive effect on the content of total polyphenols in freshly harvested fruit irrespective of the date of harvest (by 21.52% on average in relation to non-ozonated fruit), and storage time slightly decreased the parameter in red currant fruit (Table 3).

The content of compounds with antioxidant activity contained in red currant fruits depends mainly on their chemical composition, including the content of polyphenolic compounds and their differentiated structure, which affects the antioxidant potential. The antioxidant activity of the red currant fruit was determined using three methods: the DPPH radical, the ABTS cation radical, and the FRAP method. The red currant fruit harvested seven days before the optimal harvest date and not subjected to ozonation had an average antioxidant potential of 3.75 mg·mL^−1^ (DPPH), 11.49 μM TE·g^−1^ (ABTS), and 0.61 mM Fe^2+^·100 g^−1^ (FRAP). Refrigerated storage only significantly increased the oxidative activity of red currant fruit determined by the FRAP method (by 18.67% on average; Table 3). The red currant fruit harvested on the optimum harvest date, not subjected to ozonation, was characterised by a higher average of 15.28% antioxidant activity, as determined by the FRAP method, and a lower average of 11.27% antioxidant potential, as determined by the DPPH radical, while the activity determined using the ABTS cation radical showed no statistically significant changes compared with the currant fruit harvested seven days earlier (Table 3). Refrigerated storage for 15 days increased the antioxidant activity of red currant fruit, as determined by the DPPH method, by an average of 10.75%, while the antioxidant potential determined by the other methods did not show statistically significant differences for fresh fruit harvested at the optimal harvest date (Table 3). The antioxidant activity determined by the red currant DPPH method in a study by Laczkó-Zöld et al. [29] was 5.72–34.26 mg·mL^−1^ depending on the extract, while in the study by Djordjevic et al. [12], it was 1.9–12.3 mg·mL^−1^. In general, the red currant fruit ozonation process affected the variation in antioxidant activity positively compared with non-ozonated fruit. The highest average values determined by the DPPH and FRAP methods were obtained for fruit harvested 7 days before the optimal harvest date and stored for 15 days under refrigeration conditions previously ozonated for 15 min. On the other hand, red currant fruit harvested on the optimum harvest date and stored under refrigeration conditions had the highest average antioxidant activity determined using the ABTS cation radical. A similar relationship was observed for fruit treated with 10 ppm ozone gas for 30 min, and the highest antioxidant potential values were obtained for red currant fruit harvested one week before the optimal harvest date and stored for 15 days in refrigerated conditions; an average of 3.84 mg·mL^−1^ (DPPH method), 12.39 μM TE·g^−1^ (ABTS method), and 0.75 mM Fe^2+^·100 g^−1^ (FRAP method), respectively (Table 3).

### 2.4. Changes in Microbiological Properties of Ozone-Treated Red Currant Fruit

The storage life of fruit depends significantly on the content of microorganisms on the fruit surface, which activate unfavourable biochemical transformations in the fruit, resulting in the loss of the required quality. Ozone is an abiotic factor that damages the metabolism of microorganisms on the fruit, thus causing an increase in their storage life [18].

The highest number of mesophilic aerobic bacteria was recorded after one day of storage for the fruit of the control variant of the red currant cultivars studied, while the ozonated fruit showed a reduction in the number of colony-forming units of these bacteria compared with the control variant. On the date analysed, gaseous ozone at 10 ppm for 15 min reduced the concentration of mesophilic aerobic bacteria by an average of 36%, while extending the ozonation time to 30 min reduced the number of aerobic mesophilic aerobic bacteria by an average of only 27% for the varieties in relation to the control (Table 4). On day 15 of storage, we also observed that gaseous ozone had a favourable effect, reducing the number of aerobic mesophilic bacteria analysed. Compared with the results on day 1 of storage, the number of bacteria tested for each variant from the experiment increased. Over 15 days of fruit storage, ozonation for 30 min was found to have the most beneficial effect on reducing the number of aerobic colony-forming units of mesophilic bacteria. The lowest number of tested bacteria was recorded for the ‘Losan’ cultivar both on day 1 and on day 15 of storage. The red currant fruit treated with gaseous ozone for 30 min reduced the number of mesophilic bacteria by an average of 25.8% for the three cultivars tested while reducing the time to 15 min reduced the number of colony-forming aerobic bacteria by an average of 22.3% compared with the control sample (Table 4). The application of gaseous ozone during blueberry storage was effective in inhibiting the development of grey mould in the fruit tested [31]. Fumigation with low-concentration gaseous ozone helped to reduce the number of aerobic mesophilic bacteria and moulds on harvested asparagus during its storage period [32]. Similar relationships were observed for garden rhubarb. The postharvest application of gaseous ozone to rhubarb petioles reduced the number of aerobic mesophilic bacteria, as well as yeasts and moulds [17]. The application of ozone to the fruits of rhubarb resulted in a slower microflora growth rate for the ozonated variants compared with the fruits of the sample not treated with this gas [17]. An ozone concentration of 10 ppm applied for 30 min was found to reduce the number of aerobic bacteria, as well as yeast and moulds, during the storage period of Saskatoon berry fruits [33].

During storage, a high burden of yeast and mould was observed in red currant fruit. For each date of the ozonation process, irrespective of the dose applied, this had an effect on the reduction of the microbial load on the fruit. During the 15-day storage period, the results showed that the highest incidence of yeasts and moulds was on the fruit of the control trial. For the varieties treated with ozone gas after harvest, it was observed that the application of ozone gas for 15 and 30 min reduced the microbial load tested compared with the control. For these variants, the lowest values of yeast and mould infestation were observed on day 1 of storage. On day 15 of red currant fruit storage, the lowest microbial infestation was characterised for the ‘Losan’ variety ozonated after harvest for 30 min compared with the other varieties. On the basis of the analysis performed on the last day of storage, it was found that ozonation for 30 min reduced the amount of yeast and mould on the cultivars tested by an average of 1.18 log cfu·g^−1^ compared with the control sample. On the contrary, the same dose of ozone applied for 15 min reduced the number of microorganisms tested by an average of 1.04 log cfu·g^−1^ compared with the control variant (Table 5). In one study, fumigating marjoram plants with gaseous ozone resulted in a significant reduction in the number of yeasts and moulds on the first and fifth day after treatment. The most beneficial effects were observed when marjoram plants were treated with ozone for 10 min [34]. The researchers fumigated sea buckthorn berries with gaseous ozone at a concentration of 100 ppm for 30 min. The applied process conditions reduced the number of yeasts and moulds by 1 log cfu·g^−1^ after applying these conditions compared with the control. Consequently, ozone treatment improved the quality of plants and prolonged their life [18]. The ozonation of Saskatoon berry fruits had a beneficial effect on fruit quality, reducing the growth and development of yeasts and moulds during storage compared with a non-ozonated control sample [22].

## 3. Materials and Methods

### 3.1. Materials

The study materials consisted of red currant fruits of the cultivars ‘Holenderska Czerwona’, ‘Luna’, and ‘Losan’. Fruits were harvested manually in an organic farm located in Łopuszka Wielka (49°56′12″ N 22°23′35″ E, Podkarpackie Voivodeship, Poland) in the amount of 6000 g each on two harvesting dates: P—seven days before harvest maturity (first decade of July 2022) and O—at harvest maturity (second decade of July 2022). The date of harvest and the degree of ripeness of the red currant fruits were determined on the basis of their colour and the strength of binding to the stalk.

The red currant fruits, both those that were ozonated and those not subjected to this process, were stored in cold storage (temperature 3 °C) for 1, 8, and 15 days.

### 3.2. Treatment of Fruit Ozone

Immediately after harvest, the fruit was randomized into three batches of 2000 g each. The first batch was left untreated (control sample). The remaining two batches were subjected to ozonation in a plastic container, with dimensions L × W × H of 0.6 × 0.4 × 0.4 m. Gaseous ozone was used at a concentration of 10 ppm for 15 and 30 min (flow 40 g O_3_·h^−1^, temperature 20 °C). Ozone was produced with a KORONA A 40 Standard (Korona, Piotrków Trybunalski, Poland) with a 106 M UV Ozone Solution detector (Ozone Solution, Hull, MA, USA).

### 3.3. Determination of the Morphological Characteristics of Red Currant Fruits

The sample size was 15 fruits from each variant. For individual fruits, the diameter, d, was determined with an accuracy of 0.01 mm and the weight was determined with an accuracy of 0.001 g. The density (kg·m^−3^) of the individual fruits was calculated as the ratio of their weight to the volume of the sphere with diameter d [6,35].

### 3.4. Water Content Measurement

The water content of the individual tested red currant fruits was determined using the drying method (105 °C), in accordance with PN-90/A-75101-03: 1990 [36], using a laboratory moisture analyser (Radwag, Poland).

### 3.5. Determination of the Mechanical Properties of Red Currant Fruits

The selected mechanical parameters of the currant fruits were measured in a compression test between two horizontal planes using the Brookfield CT3-1000 texture analyser (AMETEK Brookfield, Middleboro, MA, USA) and using TexturePro CT software. The initial tension force of the specimen was 0.05 N, and the compression velocity was 0.2 mm·s^−1^. The destructive force, F_D_; the absolute strain, λ; and the destructive energy, E_D_, were recorded after each measurement. Relative deformation, ε, was calculated as the ratio of absolute deformation, λ, and fruit diameter, d (mm), and then expressed as a percentage [35]. The value of the apparent modulus of elasticity, E_C_, which is a measure of the effective value of the mechanical resistance of the test material, was calculated from a modified formula [6]:(1)$$E_{c}=\frac{E_{D}}{0.26\;\cdot \;d^{2}\;\cdot \;\lambda }$$
where:*E_c_*—apparent modulus of elasticity (MPa);*E_D_*—destructive energy (mJ);*d*—diameter of the fruit (mm);*λ*—deformation of the fruit in the direction of the load (mm).

### 3.6. Determination of pH and Acidity of Red Currant Fruit

The total acidity per citric acid and the pH of the red currant fruit were determined via the potentiometric titration of the analysed sample with a standard solution of 0.1 M NaOH to pH = 8.1 using a titrator (TitroLine 5000, Mainz, Germany) according to the method in PN-EN 12147:2000 [37]. Analyses were performed in 3 replicates.

### 3.7. Determination of Bioactive Components

The determination of ascorbic acid content in red currant fruit was performed according to PN-A-04019:1998 [38]. The total polyphenol content in the red currant fruit was determined using the Folin–Ciocalteu method according to the methodology described in Piechowiak et al. [14]. The free radical scavenging activity (DPPH method) was determined according to the methodology described in Djordjević et al. [12] and expressed as IC50 (mg·mL^−1^). The antioxidant activity, using the ABTS method, was determined according to the methodology described by Jakobek et al. [30]; the result is provided in μM TE·g^−1^ of fruit. The iron-reducing capacity (FRAP method) was determined according to the methodology in Chiabrando and Giacalone [39]; the result is provided in mM Fe^2+^·100 g^−1^ of fruit. All analyses were performed in triplicate.

### 3.8. Microbiological Analysis of Red Currant Fruit

The red currant fruits of the different variants of the experiment were subjected to microbiological analyses on the 1st and 15th day of storage (after prior ozonation treatment). The amount of mesophilic aerobic bacteria and the amount of yeast and mould were determined according to the methodology described by Zardzewiały et. al. [17].

### 3.9. Statistical Analysis

The Statistica 13.3. program (TIBCO Software Inc., Tulsa, OK, USA) was used to calculate a statistical evaluation of the results, which included analysis of variance (ANOVA) and the significance LSD test at a significance level of α = 0.05.

## 4. Conclusions

Organically cultivated red currant fruits harvested seven days before the optimal harvest date were characterised by the highest total polyphenol content, and cold storage decreased the parameter by 23.16% on average while also causing an increase in ascorbic acid content by 36.93% on average. The raw material harvested at harvest maturity showed significant differences in the selected parameters and was characterized by a higher fruit diameter and weight and significantly lower water content compared with the fruit harvested a week earlier, and it showed the highest antioxidant activity among the analysed cultivars (determined by the FRAP method). After 15 days of the cold storage of red currant fruit, ozonation increased the antioxidant activity, as determined by the DPPH method, by an average of 10.75%. The ozonation process also had a beneficial effect on the antioxidant potential of the red currant fruit, especially on fruit treated with 10 ppm ozone gas for 15 min. Of the red currant cultivars grown on the organic farm, the ‘Losan’ cultivar had the highest microbial stability, both for ozonated and non-ozonated fruit, regardless of harvest date. The ozonation of fruit during storage yielded better elastic properties (higher apparent modulus of elasticity). After 15 days of storage, the fruit ozonated for 30 min was characterized by higher values of energy and destructive force; i.e., they were more resistant to mechanical damage.

## Figures and Tables

**Figure 1 molecules-27-08231-f001:**
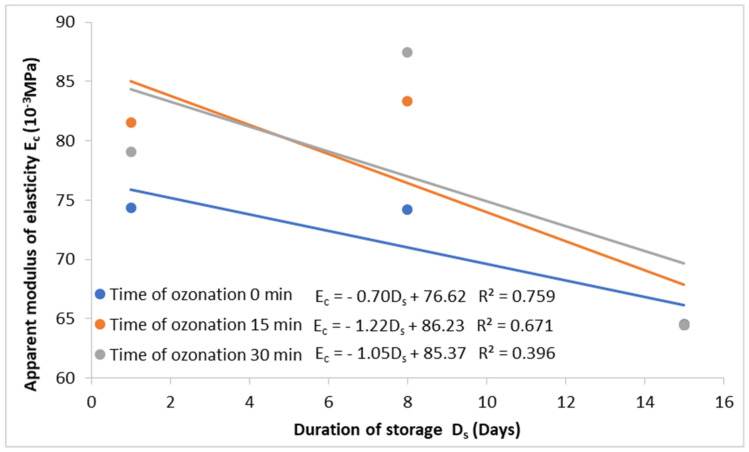
The dependence of the destructive force and energy, as well as the apparent modulus of elasticity, on the storage time of red currant fruit for three varieties and the ozonation time.

**Figure 2 molecules-27-08231-f002:**
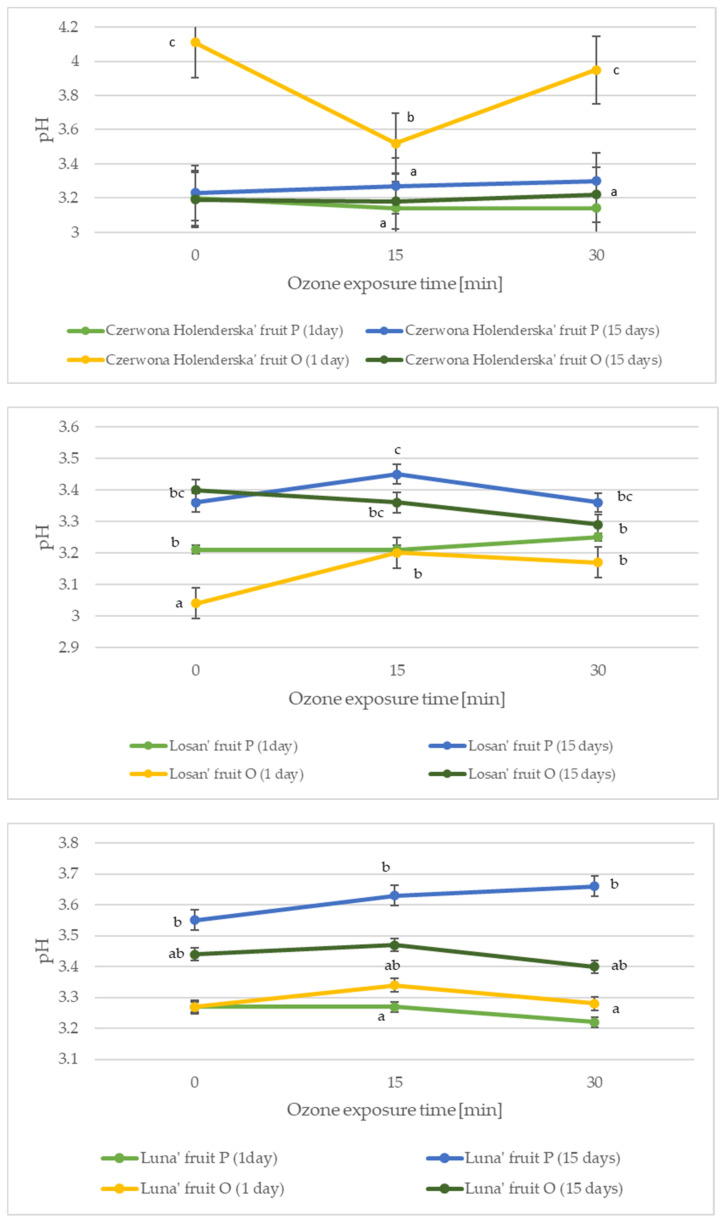
Changes in the pH of red currant fruit depending on the harvest date, variety, and ozone exposure time. Data are expressed as mean values (n = 3) ± SD; SD—standard deviation. Mean values with different letters are significantly different (*p* < 0.05).

**Figure 3 molecules-27-08231-f003:**
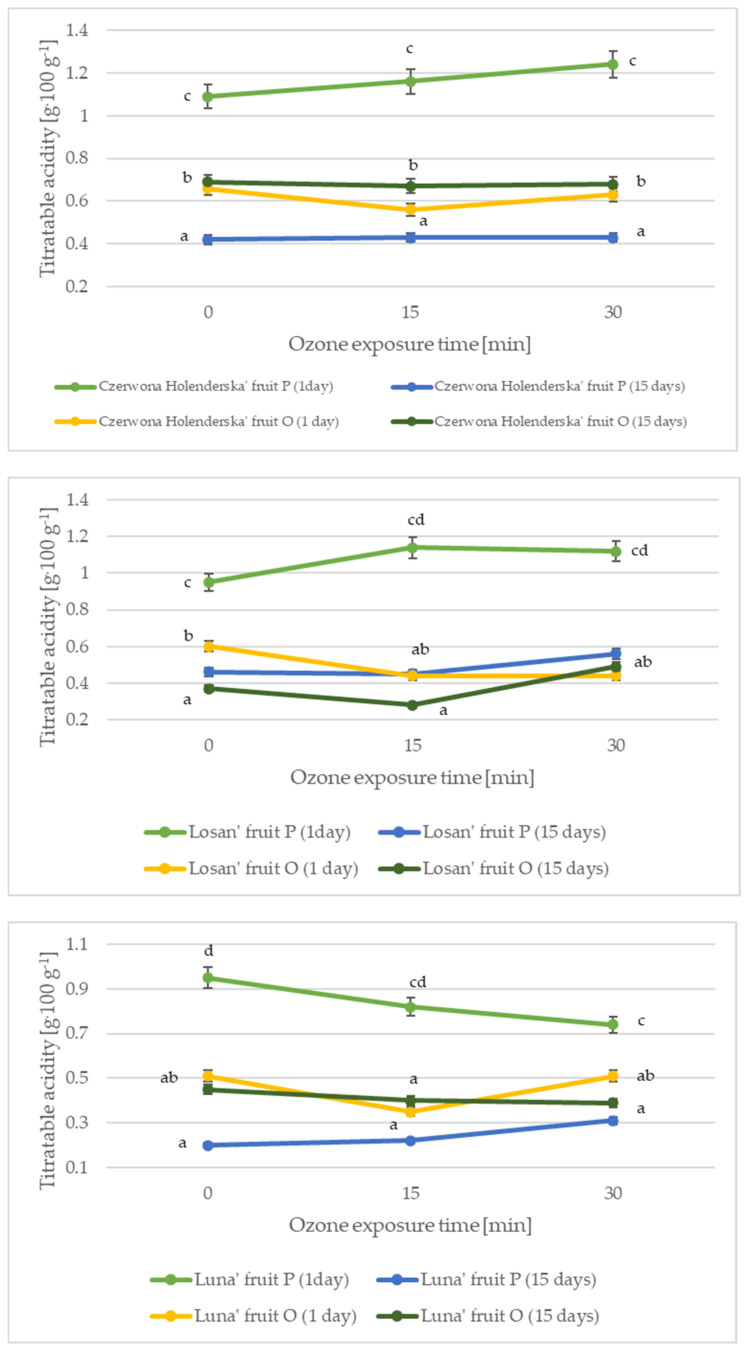
Changes in the acidity of red currant fruit depending on the harvest date, variety, and ozone exposure time. Data are expressed as mean values (n = 3) ± SD; SD—standard deviation. Mean values with different letters are significantly different (*p* < 0.05).

**Table 1 molecules-27-08231-t001:** Morphological features and moisture content of red currant fruits depending on the variety, duration of storage, and time of gaseous ozonation.

Variables		Diameter (mm)	Weight (mg)	Density (10^−3^ kg·m^−3^)	Moisture Content (%)
**Cultivar**	‘Holenderska Czerwona’	9.3 ^b^ ± 0.7	464 ^b^ ± 110	1.09 ^a^ ± 0.05	84.2 ^a^ ± 1.8
	‘Losan’	8.8 ^a^ ± 0.8	408 ^a^ ± 109	1.11 ^b^ ± 0.07	85.2 ^b^ ± 1.8
	‘Luna’	8.7 ^a^ ± 0.7	403 ^b^ ± 94	1.13 ^c^ ± 0.06	85.2 ^b^ ± 2.1
**Time of gaseous ozonation (min)**	0	9.1 ^b^ ± 0.7	448 ^b^ ± 109	1.11 ^b^ ± 0.06	85.3 ^b^ ± 2.1
	15	8.8 ^a^ ± 0.8	411 ^a^ ± 105	1.11 ^b^ ± 0.06	84.8 ^a^ ± 2.0
	30	8.9 ^a^ ± 0.8	415 ^a^ ± 106	1.12 ^b^ ± 0.07	84.6 ^a^ ± 1.8
**Duration of** **storage (days)**	1	9.1 ^b^ ± 0.8	444 ^b^ ± 110	1.12 ^b^ ± 0.07	85.4 ^b^ ± 2.1
	8	8.9 ^a^ ± 0.9	410 ^a^ ± 123	1.12 ^b^ ± 0.06	85.2 ^b^ ± 2.1
	15	8.8 ^a^ ± 0.6	420 ^a^ ± 85	1.10 ^a^ ± 0.06	84.2 ^a^ ± 1.7
**Harvest date**	P	8.9 ^a^ ± 0.7	413 ^a^ ± 105	1.11 ^b^ ± 0.06	85.1 ^b^ ± 2.0
	O	9.0 ^b^ ± 0.8	437 ^b^ ± 110	1.11 ^b^ ± 0.07	84.7 ^a^ ± 2.0
	Mean	8.9 ± 0.8	425 ± 108	1.11 ± 0.06	84.89 ± 2.0

Data are expressed as mean values (n = 10) ± SD; SD—standard deviation. Mean values within columns with different letters are significantly different (*p* < 0.05).

**Table 2 molecules-27-08231-t002:** Mechanical properties of red currant fruits depending on the variety, duration of storage, and time of gaseous ozonation.

Variables		Force (N)	Deformation (%)	Energy (mJ)	Apparent Modulus of Elasticity (10^−3^ MPa)
Cultivar	‘Holenderska Czerwona’	2.76 ^b^ ± 0.78	34.8 ^a^ ± 5.8	4.84 ^b^ ± 1.27	69.85 ^b^ ± 22.56
	‘Losan’	2.44 ^a^ ± 1.03	37.0 ^b^ ± 6.6	3.88 ^a^ ± 1.41	60.66 ^a^ ± 21.31
	‘Luna’	3.49 ^c^ ± 1.30	39.8 ^c^ ± 6.2	5.57 ^c^ ± 1.74	81.28 ^c^ ± 21.94
Time of ozonation (min)	0	3.00 ^a^ ± 1.18	37.5 ^a^ ± 6.6	5.00 ^b^ ± 1.74	68.96 ^a^ ± 22.08
	15	2.81 ^a^ ± 1.13	36.9 ^a^ ± 6.3	4.61 ^a^ ± 1.57	70.88 ^a^ ± 22.93
	30	2.88 ^a^ ± 1.12	37.1 ^a^ ± 6.6	4.69 ^a^ ± 1.58	71.94 ^a^ ± 25.32
Duration of storage (days)	1	3.11 ^b^ ± 1.13	37.4 ^a^ ± 6.3	5.46 ^c^ ± 1.70	77.65 ^c^ ± 24.46
	8	2.83 ^a^ ± 1.12	37.7 ^a^ ± 6.9	4.73 ^b^ ± 1.50	73.73 ^b^ ± 24.77
	15	2.75 ^a^ ± 1.15	36.4 ^a^ ± 6.2	4.10 ^a^ ± 1.42	60.41 ^a^ ± 16.82
Harvest date	P	2.97 ^a^ ± 1.07	37.4 ^a^ ± 6.4	4.96 ^b^ ± 1.65	74.83 ^b^ ± 23.75
	O	2.82 ^a^ ± 1.21	37.0 ^a^ ± 6.6	4.57 ^a^ ± 1.61	66.36 ^a^ ± 22.47
	Mean	2.90 ± 1.14	37.2 ± 6.5	4.76 ± 1.64	70.60 ± 23.49

Data are expressed as mean values (n = 15) ± SD; SD—standard deviation. Mean values within columns with different letters are significantly different (*p* < 0.05).

**Table 3 molecules-27-08231-t003:** The content of ascorbic acid, total polyphenols, and antioxidant activity of red currant fruit.

Cultivar	Time Gaseous Ozonation (min)	Harvest Date	Duration of Storage (Days)	Ascorbic Acid Content (mg·100 g^−1^)	Total Polyphenols Content (mg GAE·100 g^−1^)	Antioxidant Activity		
						DPPH IC_50_ (mg/mL)	ABTS^+^ (μM TE·g^−1^ d.m.)	FRAP (mM Fe^2+^ 100 g)
‘Holenderska Czerwona’	0	P	1	34.1 ^a^ ± 0.5	117.4 ^a^ ± 0.6	3.79 ^a^ ± 0.06	11.27 ^a^ ± 0.27	0.59 ^a^ ± 0.06
	0	P	15	50.2 ^b^ ± 0.6	122.6 ^a^ ± 1.4	3.86 ^a^ ± 0.12	11.52 ^b^ ± 0.09	0.80 ^b^ ± 0.10
	0	O	1	35.1 ^a^ ± 0.8	87.1 ^b^ ± 0.7	3.31 ^a^ ± 0.09	11.48 ^a^ ± 0.07	0.71 ^a^ ± 0.14
	0	O	15	40.1 ^b^ ± 0.5	79.1 ^a^ ± 1.0	3.53 ^b^ ± 0.05	11.92 ^b^ ± 0.07	0.74 ^a^ ± 0.04
	15	P	1	42.5 ^a^ ± 0.5	127.9 ^b^ ± 0.8	3.80 ^a^ ± 0.10	11.83 ^a^ ± 0.08	0.50 ^a^ ± 0.05
	15	P	15	48.2 ^b^ ± 0.2	105.1 ^a^ ± 0.6	4.06 ^b^ ± 0.06	12.02 ^b^ ± 0.07	0.75 ^b^ ± 0.07
	15	O	1	41.6 ^a^ ± 0.4	76.9 ^b^ ± 0.4	3.22 ^a^ ± 0.02	11.56 ^a^ ± 0.26	0.72 ^a^ ± 0.07
	15	O	15	42.5 ^b^ ± 0.4	43.6 ^a^ ± 0.6	3.48 ^b^ ± 0.19	12.01 ^b^ ± 0.06	0.75 ^a^ ± 0.06
	30	P	1	25.4 ^a^ ± 0.1	254.0 ^b^ ± 1.0	3.78 ^a^ ± 0.02	12.12 ^a^ ± 0.12	0.43 ^a^ ± 0.03
	30	P	15	54.3 ^b^ ± 0.3	96.2 ^a^ ± 0.2	3.89 ^a^ ± 0.07	12.31 ^b^ ± 0.08	0.74 ^b^ ± 0.04
	30	O	1	47.7 ^b^ ± 0.7	122.4 ^b^ ± 0.4	2.81 ^a^ ± 0.04	11.89 ^a^ ± 0.06	0.70 ^a^ ± 0.10
	30	O	15	38.4 ^a^ ± 0.6	83.1 ^a^ ± 0.1	3.30 ^b^ ± 0.30	12.20 ^b^ ± 0.10	0.75 ^a^ ± 0.10
‘Luna’	0	P	1	31.2 ^a^ ± 0.2	119.9 ^b^ ± 0.9	3.69 ^a^ ± 0.05	11.18 ^a^ ± 0.02	0.64 ^a^ ± 0.04
	0	P	15	55.2 ^b^ ± 0.5	83.0 ^a^ ± 1.0	3.77 ^a^ ± 0.03	11.27 ^a^ ± 0.03	0.74 ^b^ ± 0.01
	0	O	1	64.7 ^b^ ± 0.3	41.6 ^a^ ± 0.6	3.48 ^a^ ± 0.03	11.53 ^a^ ± 0.03	0.74 ^a^ ± 0.04
	0	O	15	54.9 ^a^ ± 0.1	85.8 ^b^ ± 0.6	3.83 ^b^ ± 0.03	11.85 ^b^ ± 0.05	0.75 ^a^ ± 0.06
	15	P	1	45.2 ^a^ ± 0.2	190.4 ^b^ ± 0.4	3.67 ^a^ ± 0.07	11.97 ^a^ ± 0.03	0.56 ^a^ ± 0.06
	15	P	15	59.0 ^b^ ± 1.0	63.5 ^a^ ± 0.5	3.92 ^b^ ± 0.02	12.23 ^b^ ± 0.03	0.77 ^b^ ± 0.07
	15	O	1	51.3 ^a^ ± 0.3	79.8 ^b^ ± 0.2	2.83 ^a^ ± 0.03	11.71 ^a^ ± 0.01	0.74 ^a^ ± 0.02
	15	O	15	66.0 ^b^ ± 1.0	77.1 ^a^ ± 0.1	3.46 ^b^ ± 0.06	12.67 ^b^ ± 0.07	0.74 ^a^ ± 0.03
	30	P	1	27.3 ^a^ ± 0.3	231.6 ^b^ ± 0.6	3.65 ^a^ ± 0.05	12.39 ^a^ ± 0.10	0.69 ^a^ ± 0.01
	30	P	15	60.2 ^b^ ± 0.2	79.4 ^a^ ± 0.4	3.77 ^a^ ± 0.03	12.77 ^b^ ± 0.07	0.76 ^a^ ± 0.06
	30	O	1	69.7 ^a^ ± 0.7	95.7 ^b^ ± 0.7	2.24 ^a^ ± 0.04	12.06 ^a^ ± 0.06	0.74 ^a^ ± 0.01
	30	O	15	70.1 ^a^ ± 0.1	41.7 ^a^ ± 0.7	3.92 ^b^ ± 0.02	12.62 ^b^ ± 0.02	0.75 ^a^ ± 0.03
‘Losan’	0	P	1	44.1 ^a^ ± 0.1	201.7 ^b^ ± 0.7	3.78 ^a^ ± 0.08	12.03 ^a^ ± 0.03	0.59 ^a^ ± 0.05
	0	P	15	68.2 ^b^ ± 0.2	131.7 ^a^ ± 0.7	3.78 ^a^ ± 0.02	12.31 ^b^ ± 0.08	0.70 ^b^ ± 0.10
	0	O	1	39.6 ^a^ ± 0.6	91.1 ^b^ ± 0.1	3.18 ^a^ ± 0.18	12.18 ^a^ ± 0.08	0.72 ^a^ ± 0.02
	0	O	15	43.6 ^b^ ± 0.6	46.7 ^a^ ± 0.7	3.81 ^b^ ± 0.06	12.59 ^b^ ± 0.09	0.73 ^a^ ± 0.03
	15	P	1	41.4 ^a^ ± 0.4	102.4 ^a^ ± 0.4	3.73 ^a^ ± 0.03	11.58 ^a^ ± 0.12	0.70 ^a^ ± 0.10
	15	P	15	43.3 ^b^ ± 0.3	127.3 ^b^ ± 0.3	3.74 ^a^ ± 0.04	11.83 ^b^ ± 0.03	0.73 ^a^ ± 0.03
	15	O	1	40.8 ^a^ ± 0.8	94.5 ^b^ ± 0.5	3.52 ^a^ ± 0.08	11.71 ^a^ ± 0.11	0.74 ^a^ ± 0.00
	15	O	15	45.8 ^b^ ± 0.8	67.4 ^a^ ± 0.4	3.81 ^b^ ± 0.09	11.94 ^b^ ± 0.04	0.74 ^a^ ± 0.06
	30	P	1	34.0 ^a^ ± 1.0	260.6 ^b^ ± 0.3	3.73 ^a^ ± 0.03	11.87 ^a^ ± 0.06	0.63 ^a^ ± 0.03
	30	P	15	53.2 ^b^ ± 0.2	81.1 ^a^ ± 0.1	3.86 ^a^ ± 0.04	12.06 ^b^ ± 0.06	0.74 ^b^ ± 0.04
	30	O	1	51.2 ^a^ ± 0.2	112.7 ^b^ ± 0.7	3.48 ^a^ ± 0.08	11.94 ^a^ ± 0.04	0.72 ^a^ ± 0.02
	30	O	15	49.9 ^a^ ± 0.1	85.6 ^a^ ± 0.6	3.85 ^b^ ± 0.05	12.08 ^b^ ± 0.11	0.74 ^a^ ± 0.01

Data are expressed as mean values (n = 3) ± SD; SD—standard deviation. Mean values within columns with different letters for the duration of storage are significantly different (*p* < 0.05).

**Table 4 molecules-27-08231-t004:** Microbiological load of mesophilic aerobic bacteria in red currant fruit during storage.

Cultivar	Time Gaseous Ozonation (min)	The Date of the Test			
		*p*		O	
		1 Day after Ozonation (log cfu g^−1^)	15 Days after Ozonation (log cfu g^−1^)	1 Day after Ozonation (log cfu g^−1^)	15 Days after Ozonation (log cfu g^−1^)
	0	4.71 ^b,A^	6.17 ^b,B^	5.63 ^b,A^	6.34 ^b,B^
‘Holenderska Czerwona’	15	3.95 ^b,B^	4.99 ^a,B^	3.41 ^a,A^	4.98 ^a,B^
	30	4.03 ^a,A^	4.59 ^a,A^	3.72 ^a,A^	4.78 ^a,B^
	0	3.52 ^b,A^	4.76 ^b,B^	2.81 ^b,A^	3.18 ^a,A^
‘Losan’	15	2.69 ^a,A^	3.89 ^a,B^	1.60 ^a,A^	2.59 ^a,B^
	30	3.01 ^a,A^	3.71 ^a,A^	2.18 ^a,A^	2.41 ^a,A^
	0	4.79 ^b,A^	5.02 ^a,B^	4.85 ^b,A^	5.36 ^b,A^
‘Luna’	15	3.59 ^a,B^	4.52 ^a,B^	3.32 ^a,A^	3.92 ^a,A^
	30	3.89 ^a,A^	4.31 ^a,B^	3.59 ^a,A^	3.83 ^a,A^

Data are expressed as mean values (n = 3). Different small letters denote differences in the results between ozone doses on individual days, and different capital letters indicate differences between the dates of measurements; *p* < 0.05.

**Table 5 molecules-27-08231-t005:** Microbial load of yeasts and moulds in red currant fruit during storage.

Cultivar	Ozone Exposure Time (min)	The Date of the Test			
		*p*		O	
		1 Day after Ozonation (log cfu g^−1^)	15 Days after Ozonation (log cfu g^−1^)	1 Day after Ozonation (log cfu g^−1^)	15 Days after Ozonation (log cfu g^−1^)
	0	6.01 ^b,A^	7.01 ^b,B^	5.99 ^b,A^	6.05 ^b,A^
‘Holenderska Czerwona’	15	5.40 ^a,A^	6.31 ^a,B^	5.32 ^a,A^	5.64 ^b,A^
	30	5.32 ^a,A^	6.02 ^a,B^	5.40 ^a,A^	5.54 ^a,A^
	0	4.59 ^b,A^	5.89 ^b,B^	4.62 ^b,A^	5.81 ^b,B^
‘Losan’	15	4.01 ^a,A^	4.82 ^a,B^	3.46 ^a,A^	3.54 ^a,A^
	30	3.92 ^a,A^	4.51 ^a,A^	3.18 ^a,A^	3.43 ^a,A^
	0	4.99 ^b,A^	5.61 ^a,A^	5.46 ^b,A^	5.73 ^b,A^
‘Luna’	15	4.60 ^a,A^	5.01 ^a,A^	4.32 ^b,A^	5.30 ^b,B^
	30	4.46 ^a,A^	4.82 ^a,A^	4.28 ^a,A^	5.08 ^a,B^

Data are expressed as mean values (n = 3). Different small letters denote differences in the results between ozone doses on individual days, and different capital letters indicate differences between the dates of measurements; *p* < 0.05.

## Data Availability

Not applicable.

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
