# Peer review of "Effect of Ozonation on the Mechanical, Chemical, and Microbiological Properties of Organically Grown Red Currant (Ribes rubrum L.) Fruit"

_molecules, 2022, doi:10.3390/molecules27238231_

Round 1

Reviewer 1 Report

  1. The introduction is well described, but is there anything else you can describe about the ozonation process and its impact on the products of the food industry? 
  2. 3.1. Material: How was redcurrant fruit ripeness assessed for the set harvest dates? 
  3. In lines 108-110 it is stated that "The application of ozonation decreased the values of the determined mechanical parameters of the fruits, except that the decrease was significant only for energy independent of ozonation time", while in Table 2 the average values for Force and Apperent modulus of elasticity are also marked with different letters, which indicates significant differences. 
  4. In Figure 1, commas are used as decimal characters instead of dots. 
  5. Figures 2 and 3 give erroneous descriptions of the time of harvesting and storage of red currant fruits (another description in the methodology). 

Author Response

The Authors are grateful for the contribution of the Reviewer.

  1. The introduction is well described, but is there anything else you can describe about the ozonation process and its impact on the products of the food industry? 

Answer:

More information about the effect of ozone on food products has been added in the introduction (lines 78-85).

  1. 3.1. Material: How was redcurrant fruit ripeness assessed for the set harvest dates? 

Answer:

The date of harvest and the degree of ripeness of red currant fruit were determined on the basis of their color and the strength of binding to the stalk, using the practical knowledge and experience of the red currant producer (lines 414-416).

  1. In lines 108-110 it is stated that "The application of ozonation decreased the values of the determined mechanical parameters of the fruits, except that the decrease was significant only for energy independent of ozonation time", while in Table 2 the average values for Force and Apperent modulus of elasticity are also marked with different letters, which indicates significant differences. 

Answer:

Corrections were made in table 2.

  1. In Figure 1, commas are used as decimal characters instead of dots. 

Answer:

Corrections have been made in figure 1.

  1. Figures 2 and 3 give erroneous descriptions of the time of harvesting and storage of red currant fruits (another description in the methodology). 

Answer:

Corrections have been made to figures 2 and 3.

Reviewer 2 Report

As I find the manuscript well written and the author(s) did a great effort in conducting the experiments I need to point out a few things to reject the manuscript as it has been presented. One point whith which I am not at ease is the size the experimental units - at some point (item 3.3) there is an indication of 15 fruit. This is too few for a small sized fruit (I know it can be hard have enough fruit for bigger amounts of fruit, but then, a suggestion would be repeat the experiment another time for consistency of the data). The second point is the discussion of the data. The author(s) present the result of all variables in detail and do even compare to data of the literature. But then, there is no discussion as for what reasons ozone did, for example: 1)  what is the reason for ozone promote reduce weight, water contents and diameter? Are there damages to tissues - at which point? Epidermal wax damages or whatever, but the author (s) should at least give an idea on that effect of ozone on these variables. 2) The same is true for pH and acidity. The author(s) present a good amount of comparisons to the literature and give as well percentages of increases and decreases, but then, again, no comment what is the cause of ozone on the internal metabolism (to change or not pH and acidity) of red currant. 3) And it repeats again regarding ascorbic acid - why is there an increase ascorbic acid in ozonated fruit (lines 218 and onwards)? I think the author(s) can prepare a better discussion (beyond only comparing to data from the literature) and offer a good quality paper as the data are intriguing and shall be of interest to all of those working on sanitizing treatments. 

Author Response

The Authors are grateful for the contribution of the Reviewer.

One point whith which I am not at ease is the size the experimental units - at some point (item 3.3) there is an indication of 15 fruit. This is too few for a small sized fruit (I know it can be hard have enough fruit for bigger amounts of fruit, but then, a suggestion would be repeat the experiment another time for consistency of the data). 

Answer:

The size of the experimental sample was determined on the basis of the Student's t-test.

 1)  what is the reason for ozone promote reduce weight, water contents and diameter? Are there damages to tissues - at which point? Epidermal wax damages or whatever, but the author (s) should at least give an idea on that effect of ozone on these variables.

Answer:

The dose of ozone selected on the basis of preliminary tests did not cause any visible damage to the fruit epidermis. Determination of the effect of ozone on the water loss of fruits of the analyzed redcurrant cultivars will be the subject of further research.

2) The same is true for pH and acidity. The author(s) present a good amount of comparisons to the literature and give as well percentages of increases and decreases, but then, again, no comment what is the cause of ozone on the internal metabolism (to change or not pH and acidity) of red currant. 3) And it repeats again regarding ascorbic acid - why is there an increase ascorbic acid in ozonated fruit (lines 218 and onwards)? 

Answer:

The discussion of the results was completed (lines 259 - 272 and 226-229). Further studies are planned in cooperation with cell biology on the effect of ozone gas on individual fruit tissues and their biochemical effects.

Reviewer 3 Report

overall manuscript written well.. but need some correction

line 15 remove typo "properties,,"

line 19 mentioned physical parameters in bracket

line 23, no need to mentioned method in the abstract section, better to remove

add conclusion lines at the end of abstract,

line 32 (Russia and Ukraine)

line 33 (Poland, Germany, and France)

line 53 enter space red current

line 123, remove typo et al.

line 374 to 377, provide suitable reference

go through the whole manuscript, use full form at first use.

reference prepare according to journal guidlines

Author Response

The Authors are grateful for the contribution of the Reviewer.

line 15 remove typo "properties,,"

line 19 mentioned physical parameters in bracket

line 23, no need to mentioned method in the abstract section, better to remove

add conclusion lines at the end of abstract,

line 32 (Russia and Ukraine)

line 33 (Poland, Germany, and France)

line 53 enter space red current

line 123, remove typo et al.

line 374 to 377, provide suitable reference

go through the whole manuscript, use full form at first use.

reference prepare according to journal guidlines

Answer:

All comments of the reviewer have been corrected and included in the manuscript.